# Report

# Dynamical compensation in physiological circuits

Omer Karin[1], Avital Swisa[2], Benjamin Glaser[3], Yuval Dor[2] & Uri Alon[1,*]

## Abstract

Biological systems can maintain constant steady-state output despite variation in biochemical parameters, a property known as exact adaptation. Exact adaptation is achieved using integral feedback, an engineering strategy that ensures that the output of a system robustly tracks its desired value. However, it is unclear how physiological circuits also keep their output dynamics precise —including the amplitude and response time to a changing input. Such robustness is crucial for endocrine and neuronal homeostatic circuits because they need to provide a precise dynamic response in the face of wide variation in the physiological parameters of their target tissues; how such circuits compensate their dynamics for unavoidable natural fluctuations in parameters is unknown. Here, we present a design principle that provides the desired robustness, which we call dynamical compensation (DC). We present a class of circuits that show DC by means of a nonlinear feedback loop in which the regulated variable controls the functional mass of the controlling endocrine or neuronal tissue. This mechanism applies to the control of blood glucose by insulin and explains several experimental observations on insulin resistance. We provide evidence that this mechanism may also explain compensation and organ size control in other physiological circuits.

**Keywords** calcium homeostasis; dynamical compensation; endocrine circuits; glucose homeostasis; mathematical models of disease
**Subject Categories** Metabolism; Molecular Biology of Disease; Quantitative Biology & Dynamical Systems
**Mol Syst Biol. (2016) 12: 886**

## Introduction

Homeostatic systems maintain internal variables constant in the face of external and internal perturbations. Hormones, for instance, regulate the levels of dozens of metabolites and small molecules including blood glucose, calcium, phosphate, sodium, iron, and oxygen. Transient changes in the level of a metabolite, due to a meal or a change in consumption, lead to a change in hormone secretion. The secreted hormone then acts on many remote tissues to restore the level of the metabolite back to its baseline. This feedback control can be affected by variation in the physiological parameters of the target tissues. Such variation can arise due to disease, growth, or changes in resource allocation (Kotas & Medzhitov, 2015).

Homeostatic systems thus need to show exact adaptation, the property of maintaining a constant set point for a regulated variable despite variation in system parameters. Exact adaptation has been extensively studied in biological systems (Barkai & Leibler, 1997; Alon et al, 1999; Tyson et al, 2003; Ma et al, 2009). A central engineering strategy to robustly implement exact adaptation is integral feedback (Barkai & Leibler, 1997; Yi et al, 2000; El-Samad et al, 2002). An integral feedback controller achieves exact adaptation by integrating the error of the system over time and adjusting its output accordingly (Sontag, 2003).

However, in addition to keeping a constant steady-state output, many physiological circuits also keep their output *dynamics* precise —including the amplitude and response time to a changing input. Deviations from a precise dynamic response can cause disease. These dynamics must be precise despite variations in the physiological parameters of the remote tissues targeted by the endocrine or neuronal circuit. Such physiological parameters vary over time and between people. How precise dynamics can be robust to variation in circuit parameters has rarely been explored. Standard integral feedback models cannot provide this robustness, because important parameters such as the feedback gain affect the amplitude and response time of the circuit. There is therefore a gap in understanding how endocrine and neuronal systems are able to precisely compensate their dynamics to buffer naturally occurring variations in physiological circuit parameters.

Here, we present a design principle that provides the desired robustness, which we call dynamical compensation (DC). We show that this design arises naturally in physiological systems in which the regulated variable controls the functional mass dynamics of its regulating tissue. In particular, we show that blood glucose shows DC to variation in insulin sensitivity and insulin secretion by controlling the functional mass of pancreatic beta cells. Other physiological circuits, such as calcium homeostasis, may also have the hallmark of the present DC mechanism.

1  Department of Molecular Cell Biology, Weizmann Institute of Science, Rehovot, Israel
2  Department of Developmental Biology and Cancer Research and Molecular Biology, The Institute for Medical Research Israel-Canada, The Hebrew University-Hadassah Medical School, Jerusalem, Israel
3  Endocrinology and Metabolism Service, Department of Internal Medicine, Hadassah-Hebrew University Medical Center, Jerusalem, Israel
   *Corresponding author. Tel: +972 8 934 4448; E-mail: urialon@weizmann.ac.il

# Results

### Definition of dynamical compensation

Consider a system with an input $u(t)$ and an output $y(t,s)$ such that $s > 0$ is a parameter of the system. The system is initially at steady state with $u(0) = 0$. Dynamical compensation (DC) with respect to $s$ is that for any input $u(t)$ and any (constant) $s$ the output of the system $y(t,s)$ does not depend on $s$. That is, for any $s_1$, $s_2$ and for any time-dependent input $u(t)$, $y(t,s_1) = y(t,s_2)$.

### Dynamical compensation is not guaranteed by exact adaptation

Any system that shows DC with respect to a parameter $s$ also shows exact adaptation with respect to changes in that parameter, since the output of the system with respect to $u(t) = $ constant must be the same. Exact adaptation, though, does not entail dynamical compensation. In Fig 1A–C, we demonstrate that the classic model for exact adaption, linear integral feedback, as well as other linear models such as proportional-integral feedback, do not have dynamical compensation with respect to changes in their biochemical parameters. This includes the proportional gain parameter $s$ and the integral gain $p$. Changes in these parameters lead to changes in the response time and amplitude to a given input signal.

### A simple nonlinear integral feedback model shows dynamical compensation

We propose a mechanism for DC based on known hormonal circuit reactions (Fig 1D). The basic idea is that the regulated variable $y$ controls the functional mass $Z$ of the tissue that secretes the hormone $x$ that regulates $y$. The feedback gain of $x$ is $s$, and the feedback gain of $Z$ is $p$ and the circuit input is $u(t)$. The circuit dynamic equations are as follows:

$$\dot{y} = u_0 + u(t) - sxy \qquad (1)$$

$$\dot{x} = pZy - x \qquad (2)$$

$$\dot{Z} = Z \cdot (y - y_0) \qquad (3)$$

This circuit describes nonlinear integral feedback on the output $y$. The nonlinearity of equation (3) stems from the fact that $Z$ are cells, and hence their growth equation is autocatalytic $\dot{Z} = Z\alpha$ where $\alpha$ is the growth rate. In this circuit $\alpha$ depends on the regulated variable $y$ such that growth is zero when $y = y_0$. For example, $y$ can increase the proliferation rate $\lambda_+$ and/or decrease the removal rate $\lambda_-$ of cells, such that the two rates cross at $y = y_0$ and the growth rate is $\alpha = \lambda_+ - \lambda_-$.

We claim that this circuit has DC with respect to variation in the parameters $p$, $s$. To show this, we show that the system remains invariant after transforming $x$, $Z$ to $\hat{x} = sx, \hat{Z} = psZ$. Now the equations are independent of $p$, $s$:

$$\dot{y} = u_0 + u(t) - \hat{x}y \qquad (4)$$

$$\dot{\hat{x}} = \hat{Z}y - \hat{x} \qquad (5)$$

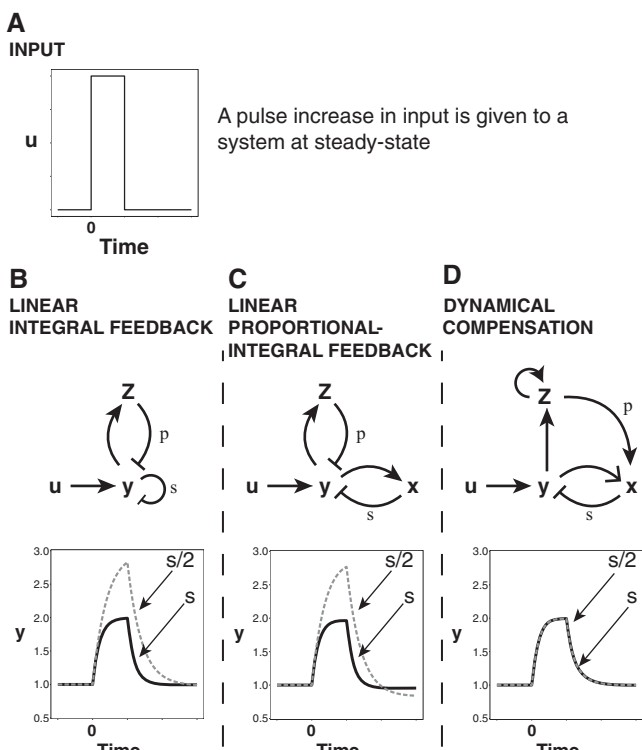

**A**
**INPUT**

A pulse increase in input is given to a system at steady-state

**B**
**LINEAR INTEGRAL FEEDBACK**

$$\dot{y} = u_0 + u(t) - sy - pZ$$
$$\dot{Z} = y - y_0$$

**C**
**LINEAR PROPORTIONAL-INTEGRAL FEEDBACK**

$$\dot{y} = u_0 + u(t) - sx - pZ$$
$$\dot{x} = y - x$$
$$\dot{Z} = y - y_0$$

**D**
**DYNAMICAL COMPENSATION**

$$\dot{y} = u_0 + u(t) - sxy$$
$$\dot{x} = pZy - x$$
$$\dot{Z} = Z \cdot (y - y_0)$$

**Figure 1.  Dynamics of input response with variation in a system parameter.**

A–D  (A) An input pulse is given to three systems: A linear integral feedback system, a linear proportional-integral feedback system, and a nonlinear feedback system with dynamical compensation. Each system has exact adaptation and thus the same steady-state output. For each system, we vary the parameter s, allow the system to reach steady state, and replot the response to the input pulse. While this changes the entire response dynamics of the linear integral controller and linear proportional-integral controller (B and C), the controller that has dynamical compensation adapts its entire dynamical trace precisely (D).

$$\dot{\hat{Z}} = \hat{Z} \cdot (y - y_0) \qquad (6)$$

The dynamics are thus independent of the parameters $p$, $s$ provided that initial conditions are also independent of $p$, $s$. To see why this is the case, consider a step change in $s:s_1 \rightarrow s_2$. Because a nonzero steady state is only possible at $y = y_0$ according to equation (6), $y$ returns to its original steady state: $y_{st} = y_0$. The steady-state levels of the scaled variables $\hat{x}$ and $\hat{Z}$ are the same before and after the change in $s$, because according to equation (1), $\hat{x}_{st} = sx_{st} = (u_0 + u(0))/y_0 = u_0/y_0$ and from equation (2): $pZ_{st} = x_{st}/y_0$ so $\hat{Z}_{st} = psZ_{st} = u_0/y_0^2$. Therefore, any input $u(t)$ will produce identical output dynamics $y(t)$ before and after $Z$ adapts to the change in the parameter $s$, because the scaled variables have the same initial conditions, and equations (4–6) depend only on the scaled variables. The same holds for a step change in $p$. Thus, the system shows DC with respect to variation in the feedback gain parameters $p$, $s$.

## Sufficient conditions for dynamical compensation

A more general class of models that show DC with respect to variations in their parameters $p$, $s$ is as follows: $\dot{y} = f(u, y, sx)$, $\dot{x} = g(y, pZ, x), \dot{Z} = Z \cdot h(y)$, provided the following sufficiency conditions: (i) For all $p$, $s$, the system is stable at $y = y^*$, there exists a unique solution $sx = x^*$ for $f(0, y^*, sx) = 0$ and there exists a unique solution $psZ = Z^*$ for $g(y^*, psZ, x^*) = 0$ (ii) A factorization condition on the function $g$: $g(y, psZ, sx) = sg(y, pZ, x)$. Proof in Appendix Section 1. An extension of this model, in which $x$ passes through multiple compartments, also shows DC with respect to variation in these parameters (Appendix Section 1).

## Glucose homeostasis shows dynamical compensation

We find the hallmarks of the DC mechanism in the well-studied glucose homeostasis system. Fasting glucose concentration in the blood is maintained within a range of about 10% around $G = 5$ mM (Allard *et al*, 2003) among healthy individuals. The glucose dynamical profile following a given glucose intake is also very similar between individuals (Ferrannini *et al*, 1985). Such constancy in dynamical profile is desirable because high plasma glucose concentrations are harmful, and too rapid a drop following a meal can cause reactive hypoglycemia which can be fatal. Accordingly, even mild abnormalities in either fasting glucose levels or postprandial glucose response dynamics are clinical indications of pre-diabetes with significant long-term health implications (Nathan *et al*, 2007).

The main regulator of plasma glucose is insulin. An increase in glucose concentration stimulates the secretion of insulin by pancreatic beta cells. Insulin acts to reduce plasma glucose levels by increasing glucose uptake in peripheral tissues and decreasing glucose production. The glucose-insulin feedback maintains glucose homeostasis on the timescale of minutes to hours.

The insulin circuit involves physiological parameters that can vary over time, primarily insulin sensitivity. Insulin sensitivity ($S_i$), the degree to which insulin is effective in lowering plasma glucose levels, can vary between individuals and throughout life by almost an order of magnitude (Bergman, 1989). Low insulin sensitivity is known as insulin resistance and is affected by obesity, inflammation, exercise, pregnancy, and genetics (Bergman, 1989).

Here, we show that glucose homeostasis has DC: The plasma glucose response to a given intake is independent of wide variations in parameters such as insulin sensitivity (Fig 2). To do so, we build on a mathematical model by Topp *et al* (2000), which is denoted as the βIG model (Fig 2A). The βIG model incorporates both the fast feedback from glucose (G) to insulin (I) as well as the long-term effect of glucose on beta-cell functional mass ($\beta$). This model has been used to explore bistability and other important aspects of glucose control (De Gaetano *et al*, 2008; Ha *et al*, 2015); here, we explore its DC property that has not been previously discussed.

In the model, plasma glucose concentration $G$ is a balance between glucose supply and removal (Bergman *et al*, 1981; Topp *et al*, 2000):

$$\dot{G} = u_0 + u(t) - (C + S_i I) \cdot G \tag{7}$$

where $I$ is plasma insulin concentration, $u_0$ is endogenous production of glucose, $u(t)$ is meal intake, $C$ is glucose removal rate at zero

insulin, and $S_i$ is insulin sensitivity. Secretion of insulin is proportional to beta-cell functional mass $\beta$ and is modeled by the equation:

$$\dot{I} = p\beta \cdot \rho(G) - \gamma I \tag{8}$$

where $\rho(G)$ is a monotonically increasing function of $G$, $\gamma$ is the insulin removal rate and $p$ is the insulin secretion per cell. So far, this model does not have DC: The glucose dynamics and steady state of equations (7) and (8) are not robust to variation in $S_i$ or any other parameter. A decrease in $S_i$ leads to an increase in steady-state glucose levels and in the meal intake response.

DC with respect to $S_i$, $p$ is achieved by an additional feedback loop in which glucose affects the production rate $\lambda_+$ and removal rate $\lambda_-$ of beta-cell functional mass. We call this feedback loop the slow feedback loop because it operates on a slower timescale than the insulin response described above. This feedback loop was first proposed by Topp *et al* (2000) based on previous experimental evidence (Swenne, 1982; Hoorens *et al*, 1996; Efanova *et al*, 1998; Hügl *et al*, 1998). The rate of change in beta-cell functional mass is:

$$\dot{\beta} = \beta(\lambda_+(G) - \lambda_-(G)) = \beta \cdot h(G) \tag{9}$$

where $h(G)$ is the net beta-cell growth rate. For this mechanism to work, we only require stability at the desired glucose set point $G = G_0$, or, in other words, $h(G_0) = 0$. Adding equation (9) makes $G$ have exact adaptation with respect to changes in $S_i$, $p$ (Topp *et al*, 2000; De Gaetano *et al*, 2008). We claim that this system also has DC with respect to $S_i$, $p$, because equations (7–9) satisfy the sufficient conditions for DC.

This means that after a change in insulin sensitivity from $S_1$ to $S_2$, beta-cell functional mass increases by a factor of $S_1/S_2$ to compensate and as a result glucose dynamics in response to a meal are precisely the same as before the change (Fig 2B). Note that the adaptation of beta-cell functional mass to the change in insulin sensitivity may take several days to months, and only after adaptation are the glucose dynamics precise. Therefore, after a step-like change in insulin sensitivity the model shows a period of time in which glucose dynamics are not fully compensated. Upon changes in insulin sensitivity that occur gradually over months, the beta cells in the model will be able to track the changes in $S_i$ and effectively compensate glucose dynamics throughout.

The DC mechanism makes additional predictions that explain experimental observations on meal responses. DC predicts that the dynamics of insulin after compensation for a change in insulin sensitivity from $S_1$ to $S_2$ will be scaled by a factor of $S_1/S_2$ (Fig 2C and D). This scaling was observed experimentally: The insulin dynamics of people with and without insulin resistance are similar when scaled by their fasting insulin level (Bagdade *et al*, 1967; Polonsky *et al*, 1988) (insets of Fig 2B–D). In fact, the steady-state solution has a constant product of insulin fasting level and insulin sensitivity ($I_{st}S_i = constant$), explaining the well-known hyperbolic relation between insulin secretion and insulin sensitivity in different individuals (Kahn *et al*, 1993).

The DC mechanism also makes the system robust to changes in plasma volume. Changes in volume rescale the concentration of $G$ and $I$, so that a ν-fold increase in plasma volume leaves equations intact except for an effective drop in insulin production per beta cell $\rho(G) \rightarrow \nu^{-1} \cdot \rho(G)$, which, just like the parameter $p$ above, is

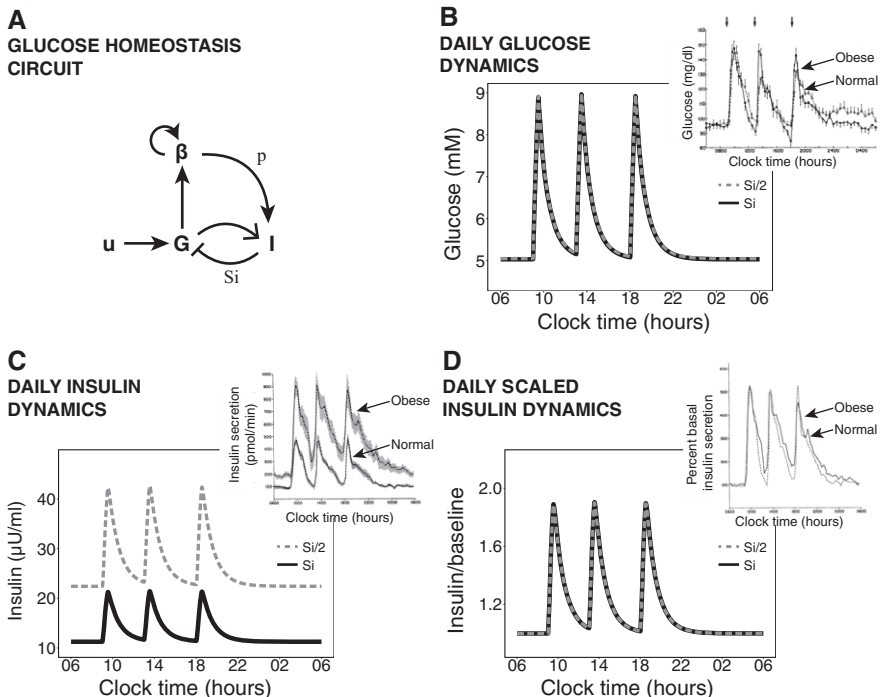

**Figure 2. Dynamical compensation after a step change in insulin sensitivity.**

A    Glucose ($G$), insulin ($I$), and beta-cell functional mass ($\beta$) interactions with a meal input ($u$). Glucose increases insulin secretion and increases beta-cell functional mass growth rate. The insulin sensitivity is $S_i$, and the insulin secretion per beta cell is $p$.

B–D    Twenty-four-hour simulated dynamics of plasma glucose concentration (B), plasma insulin concentration (C), and plasma insulin concentration normalized by its baseline (D) in response to three meals are compared before a step change in insulin sensitivity and after beta-cell adaptation to the step change. (Insets) Measured 24-h profiles of plasma glucose concentration, insulin secretion, and fold change in insulin secretion over baseline in normal and obese, insulin-resistant subjects from Polonsky et al (1988).

buffered. The DC mechanism can thus track changes in plasma volume—the higher the plasma volume, the larger beta-cell functional mass—providing a possible mechanism for the strong correlation between beta-cell mass and body weight during growth (Montanya et al, 2000). This mechanism can also guide recovery of beta-cell functional mass to the correct level following perturbations in which beta cells are lost.

We further tested whether this mechanism provides DC in a more detailed model of the insulin system by Dalla Man et al (2007). This model has 13 variables and 35 parameters. It does not include the slow feedback loop on beta-cell functional mass described here and accordingly does not show DC (Fig EV1 and Appendix Section 2). We find using numerical simulations that adding the slow feedback loop on beta-cell functional mass provides DC to changes in insulin sensitivity in this model.

For the mechanism to work, the slow feedback loop requires that beta-cell production and removal rates become equal at the glucose set-point level $G_0$. This seems to require accurate coordination between removal and production rates of beta-cells. It is well known, however, that beta-cell proliferation decreases strongly with age (Swenne, 1983), raising the question of how this coordination is achieved. We propose that the desired glucose fixed point is maintained via a switch-like drop in beta-cell removal rates around $G = G_0$. Such a sharp drop at $G = 5$ mM has been experimentally observed (Fig EV2 and Appendix Section 3).

## Pathways to failure of DC in glucose homeostasis

Despite its robustness, dynamical compensation fails in some individuals, leading to diseases such as diabetes. Diabetes is characterized by high fasting glucose and impaired glucose dynamics in response to a meal (American Diabetes Association, 2014). Diabetes can occur because of autoimmune destruction of beta cells (Type 1 Diabetes, T1D) or in a subset of individuals with insulin resistance (Type 2 Diabetes, T2D) or from other reasons. Generally, impaired glucose levels result from insulin secretion that is insufficient given the persons' sensitivity to insulin (Bergman et al, 2002).

Topp et al (2000) describe three pathways in which such insufficient secretion may develop: regulated hyperglycemia, bifurcation, and dynamical hyperglycemia. In regulated hyperglycemia, a change in beta-cell removal or production rates causes a change in the glucose set point, such that a hyperglycemic set point is maintained. In bifurcation, a more radical change may cause beta-cell removal rate to exceed their production rate at all glucose concentrations, resulting in the elimination of the beta-cell population, which may occur in T1D. The third pathway, dynamical hyperglycemia, relies on the existence of an unstable fixed point at a high glucose concentration due to the toxic effect of glucose on beta cells at these concentrations (Efanova et al, 1998). Topp et al show that in this case, if insulin sensitivity drops faster than beta-cell functional mass can adapt, then glucose levels may exceed this

unstable fixed point. In this case, the beta-cell population is eliminated. This pathway may underlie the etiology of T2D (Ha *et al*, 2015).

The above three pathways result in a perturbed glucose steady-state level. Hence, one of the conditions for DC is not met, condition (i) (stability at the desired set point). We would like to add two other mechanisms for pathology that can occur even when the normal glucose set point is maintained. First, note that a circuit with DC is not robust to all of its parameters, only to certain ones. The glucose homeostasis model has DC to the insulin sensitivity and insulin secretion parameters, which vary over a wide range. The model does not have, by itself, DC to variation in endogenous glucose production or insulin removal rate, which may vary less. Changes in these parameters alter glucose dynamics in response to a given input (Fig EV3 and Appendix Section 4). DC in the model is also affected by a mismatch between muscle and liver insulin resistance, which can alter glucose dynamics in a way that agrees with clinical observations (Fig EV4 and Appendix Section 5).

## Dynamical compensation may occur in other physiological circuits

Here, we ask whether other physiological circuits also have the DC property. In Table 1, we list several hormonal and neuronal systems that have the regulatory hallmarks for a DC mechanism. While experimental data on whether these systems indeed show dynamical compensation is lacking, we hypothesize that they are good candidates for DC.

One system that we hypothesize may show DC is calcium homeostasis. Plasma calcium levels are maintained within a tight range by the parathyroid hormone (PTH). PTH acts to increase plasma calcium levels by increasing its release from the bones, its reabsorption in the kidneys and its absorption from the intestine. The parathyroid (PT) gland secretes PTH in response to low calcium. In addition, calcium has a direct effect on PT gland mass dynamics by means of suppressing PT cell growth (Naveh-Many *et al*, 1995; Wada *et al*, 1997; Mizobuchi *et al*, 2007), forming a candidate slow feedback loop. When the PT hormone becomes less effective (for instance due to chronic renal failure), hyperplasia of the PT glands and increased secretion of PTH develops (Fukagawa *et al*, 1991). In Fig EV5 and Appendix Section 6, we show that a plausible model for the calcium homeostasis system that incorporates this slow feedback loop has DC. This model demonstrates that DC can occur also

when the hormone acts to increase the regulated variable (calcium), and not only when it acts to decrease it as in the case of glucose/insulin.

A second putative example is the control of adrenal and thyroid gland sizes by their respective trophic hormones, the regulated variables in this case, which can potentially provide dynamical compensation for stress and metabolic rate, respectively (Appendix Section 7).

A putative example for a neuronal circuit is the control of arterial oxygen by signals sent from the carotid body to increase respiration rate in response to hypoxia. Hypoxia has been shown, in turn, to increase cell proliferation in the carotid body (Nurse & Vollmer, 1997; Wang *et al*, 2008; Platero-Luengo *et al*, 2014), causing an increased ventilatory response (Teppema & Dahan, 2010; Bishop *et al*, 2013).

In these systems, the regulated variable feeds back on the size or functional mass of the regulating tissue, providing the interactions needed for a DC mechanism. The present mechanism may thus explain how organ size control of hormone-secreting glands is achieved and how hormones can function precisely despite variation in the physiological parameters of their target tissues.

## Discussion

This study presented the concept of dynamical compensation. Dynamical compensation is a property of systems in which for every possible time varying input, the complete dynamics of the output, including its amplitude and response time, are insensitive to variations in key parameters of the system. Dynamical compensation is achieved by a component (such as hormone-secreting tissue) whose functional mass changes to buffer the variation in these parameters. This property entails exact adaptation but is distinct from it, because exact adaptation only requires that the steady-state output will be robust (and not the response amplitude, response time, etc.). Dynamical compensation is especially important in systems in which improper response to input perturbations may lead to pathology, such as in metabolite homeostasis.

The concept of dynamical compensation relates to the concept of fold-change detection (FCD), in which the output dynamics of a system is independent of multiplying its input by a scalar (Shoval *et al*, 2010). Like FCD, which is an invariance property (Shoval *et al*, 2011) with respect to scaling of the input, dynamical compensation is also an invariance property, but with respect to changes in certain parameters. DC, however, is different from FCD because most known FCD mechanisms do not have dynamical compensation when their parameters are changed. Likewise, DC systems need not have FCD.

Our analysis of circuits with the DC property focused on circuits with three nodes. A three-node circuit architecture allows for a fast feedback component together with a slower nonlinear integral feedback component, which may correspond in physiological systems to a hormone and a hormone-secreting tissue, respectively. Future work may explore dynamical compensation in more complex regulatory networks.

This study provides a class of circuits that show dynamical compensation to key parameters that naturally vary over time and between people. In particular, we found that this class of circuits includes plausible models of glucose homeostasis. We analyzed a

**Table 1. Physiological systems that have the hallmarks of a DC mechanism.**

| Regulated variable (*y*) | Fast feedback (*x*) | Tissue (*Z*) |
|---|---|---|
| Plasma glucose | Insulin | Pancreatic beta cells |
| Plasma calcium | Parathyroid hormone (PTH) | Parathyroid gland |
| Arterial oxygen | Ventilatory reflex | Peripheral chemoreceptors (carotid and aortic bodies) |
| Adrenocorticotropic hormone (ACTH) | Cortisol | Adrenal gland |
| Thyroid stimulating hormone (TSH) | Thyroid hormones (T3, T4) | Thyroid gland |

model based on the work of Topp *et al* (2000). This model extends the classic glucose-insulin system by incorporating the effect of glucose on the dynamics of beta-cell mass and is used to analyze the pathogenesis of diabetes (Topp *et al*, 2000; De Gaetano *et al*, 2008; Ha *et al*, 2015). While the original model explicitly referred to the action of glucose on beta-cell proliferation and apoptosis, our analysis applies generally to removal and production of beta-cell functional mass. Removal of beta-cell functional mass can be due to dysfunction, hypotrophy, cell death or de-differentiation, and production might be due to increased function, hypertrophy, or proliferation. DC seems to occur in experimental measurements on the glucose and insulin responses of people with and without insulin resistance (Fig 2, insets). These studies reported population averages of the dynamics, which can potentially mask variations between people; data on individual dynamics would provide a more stringent test of DC. Finally, we suggest that DC may arise naturally in other physiological homeostatic circuits in which the regulated variable controls the mass dynamics of its regulating tissue.

## Materials and Methods

### βIG model parameters

For the simulations of the βIG model, presented in equations (7–9) in the main text, we used the parameters from Topp *et al* (2000) displayed in Table 2. The model is provided in SBML format in Code EV1.

To simulate equation 9 (see above), we used the following equations for the production and removal of beta-cell mass. We assumed that functional beta-cell production follows the kinetics of glucokinase (Porat *et al*, 2011):

$$\lambda_+(G) = \mu_+ \cdot \frac{1}{1 + \left(\frac{8.4}{G}\right)^{1.7}}$$

and functional beta-cell removal has the experimentally observed steep drop near 5 mM described in Appendix Section 2:

$$\lambda_+(G) = \mu_- \cdot \frac{1}{1 + \left(\frac{G}{4.8}\right)^{8.5}}$$

The values of $\mu_+$, $\mu_-$ determine the turnover of functional mass, which is less than 1% per day:

$$\mu_+ = 0.021 \cdot \frac{1}{24 \cdot 60} \ \text{min}^{-1}$$

$$\mu_- = 0.025 \cdot \frac{1}{24 \cdot 60} \ \text{min}^{-1}$$

This feedback loop, with these parameters, is also added to the model by Dalla Man *et al* (2007) (Fig EV1) to simulate the slow feedback from glucose to functional beta-cell mass.

The simulation that incorporates both hepatic and muscle insulin resistance (Fig EV4) uses normal insulin sensitivity parameters from Visentin *et al* (2015) displayed in Table 3 with all other parameters as in the βIG model.

**Expanded View** for this article is available online.

### Acknowledgements
We thank members of the Alon laboratory for discussions. UA is the incumbent of the Abisch-Frenkel Professorial Chair. OK is supported by the Azrieli Center for Systems Biology grant.

### Author contributions
OK and UA conceived and performed the research. OK and UA wrote the paper. YD, BG, and AS provided guidance on glucose homeostasis.

### Conflict of interest
The authors declare that they have no conflict of interest.

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

**Table 2. βIG model parameter values.**

| Parameter | Value | Units |
|---|---|---|
| $u_0$ | $\frac{1}{30}$ | mM min$^{-1}$ |
| $C$ | $10^{-3}$ | min$^{-1}$ |
| $S_i$ | $5 \times 10^{-4}$ | ml μU$^{-1}$ min$^{-1}$ |
| $p$ | 0.03 | μU ml$^{-1}$ min$^{-1}$ |
| $\rho(G) = \frac{G^2}{\alpha^2 + G^2}$ | $\alpha = 7.85$ | mM |
| $\gamma$ | 0.3 | min$^{-1}$ |

**Table 3. Hepatic and muscle insulin sensitivity parameter values.**

| Parameter | Value | Units |
|---|---|---|
| $S_H$ | $2.74 \times 10^{-3}$ | mM ml μU$^{-1}$ min$^{-1}$ |
| $S_M$ | $9.93 \times 10^{-4}$ | ml μU$^{-1}$ min$^{-1}$ |

Bergman RN, Phillips LS, Cobelli C (1981) Physiologic evaluation of factors controlling glucose tolerance in man: measurement of insulin sensitivity and beta-cell glucose sensitivity from the response to intravenous glucose. *J Clin Invest* 68: 1456–1467

Bishop T, Talbot NP, Turner PJ, Nicholls LG, Pascual A, Hodson EJ, Douglas G, Fielding JW, Smith TG, Demetriades M, Schofield CJ, Robbins PA, Pugh CW, Buckler KJ, Ratcliffe PJ (2013) Carotid body hyperplasia and enhanced ventilatory responses to hypoxia in mice with heterozygous deficiency of PHD2. *J Physiol* 591: 3565–3577

Dalla Man C, Rizza RA, Cobelli C (2007) Meal simulation model of the glucose-insulin system. *IEEE Trans Biomed Eng* 54: 1740–1749

De Gaetano A, Hardy T, Beck B, Abu-Raddad E, Palumbo P, Bue-Valleskey J, Porksen N (2008) Mathematical models of diabetes progression. *Am J Physiol Endocrinol Metab* 295: E1462–E1479

Efanova IB, Zaitsev SV, Zhivotovsky B, Köhler M, Efendić S, Orrenius S, Berggren PO (1998) Glucose and tolbutamide induce apoptosis in pancreatic beta-cells. A process dependent on intracellular $Ca^{2+}$ concentration. *J Biol Chem* 273: 33501–33507

El-Samad H, Goff JP, Khammash M (2002) Calcium homeostasis and parturient hypocalcemia: an integral feedback perspective. *J Theor Biol* 214: 17–29

Ferrannini E, Bjorkman O, Reichard GA, Pilo A, Olsson M, Wahren J, DeFronzo RA (1985) The disposal of an oral glucose load in healthy subjects: a quantitative study. *Diabetes* 34: 580–588

Fukagawa M, Kaname S, Igarashi T, Ogata E, Kurokawa K (1991) Regulation of parathyroid hormone synthesis in chronic renal failure in rats. *Kidney Int* 39: 874–881

Ha J, Satin LS, Sherman AS (2015) A Mathematical Model of the Pathogenesis, Prevention and Reversal of Type 2 Diabetes. *Endocrinology* 157: 624–635

Hoorens A, Van de Casteele M, Klöppel G, Pipeleers D (1996) Glucose promotes survival of rat pancreatic beta cells by activating synthesis of proteins which suppress a constitutive apoptotic program. *J Clin Invest* 98: 1568–1574

Hügl SR, White MF, Rhodes CJ (1998) Insulin-like growth factor I (IGF-I)-stimulated pancreatic beta-cell growth is glucose-dependent. Synergistic activation of insulin receptor substrate-mediated signal transduction pathways by glucose and IGF-I in INS-1 cells. *J Biol Chem* 273: 17771–17779

Kahn SE, Prigeon RL, McCulloch DK, Boyko EJ, Bergman RN, Schwartz MW, Neifing JL, Ward WK, Beard JC, Palmer JP (1993) Quantification of the relationship between insulin sensitivity and beta-cell function in human subjects. Evidence for a hyperbolic function. *Diabetes* 42: 1663–1672

Kotas ME, Medzhitov R (2015) Homeostasis, inflammation, and disease susceptibility. *Cell* 160: 816–827

Ma W, Trusina A, El-Samad H, Lim WA, Tang C (2009) Defining network topologies that can achieve biochemical adaptation. *Cell* 138: 760–773

Mizobuchi M, Ogata H, Hatamura I, Saji F, Koiwa F, Kinugasa E, Koshikawa S, Akizawa T (2007) Activation of calcium-sensing receptor accelerates apoptosis in hyperplastic parathyroid cells. *Biochem Biophys Res Commun* 362: 11–16

Montanya E, Nacher V, Biarnés M, Soler J (2000) Linear correlation between beta-cell mass and body weight throughout the lifespan in Lewis rats: role of beta-cell hyperplasia and hypertrophy. *Diabetes* 49: 1341–1346

Nathan DM, Davidson MB, DeFronzo RA, Heine RJ, Henry RR, Pratley R, Zinman B; American Diabetes Association (2007) Impaired fasting glucose and impaired glucose tolerance: implications for care. *Diabetes Care* 30: 753–759

Naveh-Many T, Rahamimov R, Livni N, Silver J (1995) Parathyroid cell proliferation in normal and chronic renal failure rats. The effects of calcium, phosphate, and vitamin D. *J Clin Invest* 96: 1786–1793

Nurse CA, Vollmer C (1997) Role of basic FGF and oxygen in control of proliferation, survival, and neuronal differentiation in carotid body chromaffin cells. *Dev Biol* 184: 197–206

Platero-Luengo A, González-Granero S, Durán R, Díaz-Castro B, Piruat JI, García-Verdugo JM, Pardal R, López-Barneo J (2014) An $O_2$-sensitive glomus cell-stem cell synapse induces carotid body growth in chronic hypoxia. *Cell* 156: 291–303

Polonsky KS, Given BD, Van Cauter E (1988) Twenty-four-hour profiles and pulsatile patterns of insulin secretion in normal and obese subjects. *J Clin Invest* 81: 442–448

Porat S, Weinberg-Corem N, Tornovsky-Babaey S, Schyr-Ben-Haroush R, Hija A, Stolovich-Rain M, Dadon D, Granot Z, Ben-Hur V, White P, Girard CA, Karni R, Kaestner KH, Ashcroft FM, Magnuson MA, Saada A, Grimsby J, Glaser B, Dor Y (2011) Control of pancreatic β cell regeneration by glucose metabolism. *Cell Metab* 13: 440–449

Shoval O, Alon U, Sontag E (2011) Symmetry invariance for adapting biological systems. *SIAM J Appl Dyn Syst* 10: 857–886

Shoval O, Goentoro L, Hart Y, Mayo A, Sontag E, Alon U (2010) Fold-change detection and scalar symmetry of sensory input fields. *Proc Natl Acad Sci USA* 107: 15995–16000

Sontag ED (2003) Adaptation and regulation with signal detection implies internal model. *Syst Control Lett* 50: 119–126

Swenne I (1982) The role of glucose in the in vitro regulation of cell cycle kinetics and proliferation of fetal pancreatic B-cells. *Diabetes* 31: 754–760

Swenne I (1983) Effects of aging on the regenerative capacity of the pancreatic B-cell of the rat. *Diabetes* 32: 14–19

Teppema LJ, Dahan A (2010) The ventilatory response to hypoxia in mammals: mechanisms, measurement, and analysis. *Physiol Rev* 90: 675–754

Topp B, Promislow K, deVries G, Miura RM, Finegood DT (2000) A model of beta-cell mass, insulin, and glucose kinetics: pathways to diabetes. *J Theor Biol* 206: 605–619

Tyson JJ, Chen KC, Novak B (2003) Sniffers, buzzers, toggles and blinkers: dynamics of regulatory and signaling pathways in the cell. *Curr Opin Cell Biol* 15: 221–231

Visentin R, Dalla Man C, Basu R, Basu A, Rizza RA, Cobelli C (2015) Hepatic insulin sensitivity in healthy and prediabetic subjects: from a dual- to a single-tracer oral minimal model. *Am J Physiol - Endocrinol Metab* 309: E161–E167

Wada M, Furuya Y, Sakiyama J, Kobayashi N, Miyata S, Ishii H, Nagano N (1997) The calcimimetic compound NPS R-568 suppresses parathyroid cell proliferation in rats with renal insufficiency. Control of parathyroid cell growth via a calcium receptor. *J Clin Invest* 100: 2977–2983

Wang Z-Y, Olson EB, Bjorling DE, Mitchell GS, Bisgard GE (2008) Sustained hypoxia-induced proliferation of carotid body type I cells in rats. *J Appl Physiol* 104: 803–808

Yi T-M, Huang Y, Simon MI, Doyle J (2000) Robust perfect adaptation in bacterial chemotaxis through integral feedback control. *Proc Natl Acad Sci USA* 97: 4649–4653

