## [Review Process File · Molecular Systems Biology]

Dynamical compensation in physiological circuits

Omer Karin, Avital Swisa, Benjamin Glaser, Yuval Dor and Uri Alon

Corresponding author: Uri Alon, Weizmann Institute of Science

Review timeline:

Submission date:	21 July 2016
Editorial Decision:	05 September 2016
Revision received:	22 September 2016
Accepted:	05 October 2016

Editor: Thomas Lemberger

Transaction Report:

1st Editorial Decision

05 September 2016

Dear Mr Karin,

Thank you again for submitting your work to Molecular Systems Biology. We have now heard back from the three referees who agreed to evaluate your manuscript. As you will see from the reports below, the referees find the topic of your study of potential interest. They raise, however, several points that we would ask you to address convincingly in a revision of this work.

The reviewers and the editor appreciate the novelty of the concept proposed in this theoretical study and the resulting insights into mechanisms of homeostasis. The reviewers point however to several aspects that deserve further clarification and modifications. One of the important points in the reports below is to clarify the potential relationship or impact of a failure in 'dynamical compensation' (this nomenclature seems actually appropriate to us) and loss of homeostasis in disease such as diabetes. Another important point is to make a clear distinction between speculative statements, which can nevertheless be interesting, from claims that are supported by (existing) data. Please note that we have also circulated the reports to each referees for cross-commenting. We received feedback from referee #1 that we include below ('cross-commenting remarks').

REFeree REPORTS

Reviewer #1:

Summary

This paper presents an interesting new viewpoint on how biological systems maintain homeostasis. The novel aspect is the proposal of a class of mechanisms that preserve the temporal characteristics of, say, the response to a pulsed challenge, not just the steady state. They show that standard integral

feedback models preserve the latter but not the former, which they term dynamic compensation (DC).

The main application is to glucose homeostasis, and the authors show that an existing class of models developed with that specific application in mind satisfy DC. They also describe more briefly other endocrine systems that have this property, supporting the idea that this is a general principle. Each of these systems responds to short-term fluctuations in the environment by negative feedback and additionally can adapt to chronic failure to meet the target by expanding the mass of endocrine tissue involved.

General Remarks

The work represents a conceptual advance in our understanding of homeostasis, which is not completely characterized by the existence of negative feedback. Although the other models of glucose-insulin homeostasis cited exhibit DC, this feature has not previously been identified properly or discussed, nor has the general principle shared with other endocrine systems been appreciated. This work should be of interest to theorists but also anyone who recognizes the remarkable robustness of homeostatic systems.

Major Points

l 250: "a circuit with DC is not robust to all parameters": This is a hint that homeostatic failure, leading to diseases such as diabetes, is possible even in systems with DC and would be an opportune place to comment briefly on a key difference between the proposed glucose-insulin model and previous models based on that of Topp et al. The authors' model exhibits DC with respect to insulin sensitivity (S_i), whereas the Topp model can fail to maintain glucose homeostasis if S_i decreases too greatly or too rapidly. This difference results from the choice of functions defining the growth and decay of beta-cell mass, which allows a unique steady state (Supplement, Materials and Methods, l 184 - l 188; Supplementary Figure 2). Whereas the Topp model allows two steady states, one stable, corresponding to successful homeostasis, and one unstable, which acts as a threshold beyond which mass collapses to a degenerate steady state of 0 mass, common to both models. Because of this, the Topp model exhibits DC provided the stress on the system is not too great but fails if a tipping point is reached.

Minor Points

l 101: the result $\hat{x}_{st} = u_0$ assumes that $u(t)$ is 0

l 112: g has an extra argument, 0

l 142: The beta-I-G model: this name has not been defined

Cross-commenting remarks:

I don't agree with Reviewer #2's insistence that the concept of dynamical compensation needs to be demonstrated with a more realistic or complex model. That would be tantamount to saying that Newton's law of gravity was not valid because it was based on observations of the oversimplified case of a falling apple instead of, say, the flight of a bird. This review is a classic case of "you didn't write the paper I would have written." It is reasonable to ask the authors to clarify how their new concept compares to other related ideas, such as fold detection, but it is not reasonable to ask them to set aside their findings until they complete a research program that could take years.

Reviewer #3 is confused about the simulations in the paper. The authors have not allowed beta cell mass to change during the course of an OGTT; they have done what he suggests they should do. This is a minor point that the authors can address easily in their rebuttal.

I do agree with the requests of both reviewers for more complete specification of the models used, parameters, etc. I stand by my recommendation that the paper can be made acceptable by minor

revisions.

Reviewer #2:

Review of the paper MSB-16-7216: "Dynamical compensation in physiological circuits"

Summary

The paper introduces the concept of Dynamical Compensation (DC), a seemingly novel property of some negative feedback loops involving an integral action (or what the authors call a slow feedback). In addition to the perfect adaptation property with respect to a given input stimulus, the dynamical compensation property ensures that, for some given input stimulus, the regulated variable will have the same response independently of the value of certain network parameters. It is a very strong property that can be understood as an invariance property with respect to some network parameters which is reminiscent of the fold-change detection property [1], an invariance property with respect to constant scaling of constant inputs. Numerical comparisons (on not necessarily physiologically realistic models) between different integral feedback strategies illustrate that this property is not ensured by every integral feedback motifs. A result stating simple sufficient conditions for a 3-node network (involving an integral feedback) to exhibit this property is also proved. Based on work reported in the literature, the authors show that a simple model of glucose homeostasis possesses such a property, and they exploit it to make predictions that seem to be validated by some existing experimental data. Finally, the authors extrapolate this result to a more complex glucose homeostasis system and to other physiological models (such as calcium regulation), this time with little supporting evidence.

General remarks

The paper is essentially of theoretical nature, as all the experimental data are taken from the existing literature in the field. Also, the paper makes strong claims that are not supported by any experimental data or experimental evidence. Nevertheless, the theoretical concept of dynamical compensation is very interesting and should have been investigated in more detail. Indeed, the results are very simple and lack generality, as they only apply to toy models. Also the connection with invariance should have been made clearer and studied in more detail using, for instance, similar ideas as in [1]. Indeed, even if fold-change detection is a different phenomenon, they are both invariance properties. In my opinion, the name "dynamical compensation" is not very suitable, as it does not properly describe the reported property. "Dynamical invariance" or "parametric invariance" seem more appropriate. In spite of this, this work nicely complements existing work pertaining to the integral control of reaction networks (such as [1]) which have clearly missed this property. Overall, I believe the paper makes a nice conceptual advance and may be publishable, but only after major revisions have been made.

Major points

1 One of the main concerns with this work is its lack of generality. First of all, the authors only consider 3-node topologies (as e.g. in [2]), which is quite restrictive. The first question that comes to mind is whether this property may hold for larger networks and, if so, how can we check it? Another limitation (which is also present in [2]) is that the input stimulus is only applied to one specific node, but what would happen if it were applied to another one? Technically, input stimuli may perturb the network at every possible node and reaction.

2 The different illustrative toy models have been often badly chosen. Indeed, some of their states can go negative, which is not very realistic from a physiological viewpoint. For instance, on page 3, a much better model could have been used than the one considered. Indeed, the model

$$\begin{aligned} y'(t) &= u_0 + u(t) - sx(t)y(t) \\ x'(t) &= pZ(t) - x(t) \quad (1) \\ Z(t) &= Z(t)(y(t) - y_0) \end{aligned}$$

would also have the DC property, would be simpler and would have its state confined to the nonnegative orthant. Additionally, the fact that the term pyZ has been changed to pZ makes the

model more interesting since x now represents the variable that will, in turn, activate the degradation of the controlled variable y . Note also that neither this model nor the original one does include proportional-integral feedback. These models only involve an integral one since the dynamics of x cannot be considered as a proportional action. It is simply low-pass filtering of the integral action. The models given in Figure 1 should also be modified so that they become physiologically more relevant. Note that this modification is possible while still keeping the main point of the authors: that not all networks with integral feedback exhibit the DC property.

3. Another point that needs to be addressed is the extent to which this property is robust. From a biological viewpoint, fragile functions or properties are not achievable. A discussion is necessary to explain how robust is the DC property. In other words, when a model is assumed to satisfy the DC property, to what extent a perturbed version of that model also satisfies that property? For instance, if a degradation term is added to the integrator (for instance, the term $-\delta Z$ where $\delta > 0$), then the adaptation property as well as the DC property are lost. These properties are non-robust with respect to perturbations at the integrator level, which is not surprising. Robustness is also essential to consider in order to capture the fact that biological systems are not isolated from each other but interfere in sometimes very strong ways. In the present case, the α -cells also act on the glucose concentration. In summary, the robustness of this property and that of the glucose regulation system should be addressed and properly discussed.

4. Page 8. It is claimed in the main text that by adding a slow feedback in the model taken from Dalla Man et al., 2007, the model then exhibits DC. This should be clearly developed in the supplementary material. In the current version of the paper, nothing is explained or even proved. The model needs to be included for completeness, and the conditions of the main result should be applied to it in order to prove the DC property.

5. The authors assume in the calcium regulation the presence of a slow feedback implementing an integral action where the functional mass of the parathyroid gland is modeled as $M = M \cdot h([Ca^{2+}])$. However, no evidence is given that this is indeed the case. The authors do not cite any experimental paper that supports such a model. Moreover, dynamical compensation does not seem to have been reported in the case of calcium homeostasis. Thus, this example seems quite contrived. One can always "fix" a model in order to have some desired properties, but this can only be considered as speculative.

6. Regarding Section 5 of the SI, the glucose dynamics can go negative and this should be corrected. The model also needs better justification.

7. All the figures in the supplementary material need additional explanations/interpretations in the text or in their legend. In particular, Figure 2a and Figure 4 need further explanations.

8. In Figure 2, the experimental curves are averaged over several individuals, a procedure that both eliminates the randomness in the trajectories and the fact that these trajectories can be very different from each other. Nothing is said about this. A consequence is that fitting a model on these average trajectories may mean nothing since the deterministic model would represent an individual while the used data is a population average. Hence, the considered model seems to be more a phenomenological model than a mechanistic one, which goes against all the arguments used in the paper to justify the structure of the models. This should be clarified.

9. It is said in the discussion section that DC is different from FCD. While this is true, the result presented in the paper is also an invariance property and should have been addressed as such using the same ideas as in [1]. It is important that the paper is rewritten to account for this similarity.

Minor points

1. In the notations, it is important to make the distinction between initial conditions (with subscript 0, * like x_0) and steady-state values with superscript * for instance like x^* . For instance, in the integrator * expression, y should be used.

1 Page 3. The definition of DC is very sloppy. It should be formulated in more rigorous way (i.e. in a mathematically rigorous way) with a possible explanation that could be similar to the one given in the present version.

2 Page 5. The statements of the conditions should also be reformulated in a less sloppy way. For instance, the uniqueness of the equilibrium point and the the local stability could be stated together. The homogeneity property can be stated on a separate item.

3 The statement and the proof of the main theoretical result need improvement. The first statement should be about the existence, uniqueness, and stability of the equilibrium point for all $s, p > 0$. The second statement is the factorization condition. Note that homogeneity is usually defined in the case of a scaling factor that applies to all the arguments of a function. This is not the case here. Hence, I would recommend to change the term to avoid confusion. The term factorization seems more appropriate. In the proof, notation using equal signs in the arguments of a function should be avoided.

4 It should be discussed how this property generalizes to larger models (more than 3 nodes) and to other stimuli, like sudden change in the parameters of the system. For instance, what happens if s or p suddenly change while u is kept constant?

5 It seems that this property crucially depends on structure of the integral action. What would happen in the presence of a small leakage in the dynamics of Z or the consideration of another integral control structure?

References

- [1] O. Shoval, U. Alon, and E. Sontag, "Symmetry invariance for adapting biological systems," *SIAM Journal on Applied Dynamical Systems*, vol. 10(3), pp. 857-886, 2011.
- [2] W. Ma, A. Trusina, H. El-Samad, W. A. Lim, and C. Tang, "Defining network topologies that can achieve biochemical adaptation," *Cell*, vol. 138, pp. 760-773, 2009.

Reviewer #3:

In this study the authors introduce the concept of dynamical compensation and explain its biomedical relevance using hormone regulatory circuits as examples. Dynamical compensation describes the ability of a biological system to respond to the same input always with the same dynamical output regardless of the actual states of some of its parameters. The authors explain and provide proof the concept first by using a simple example of a dynamical model that contains the necessary network motif, followed by a generalized model description and proof. To demonstrate the biological/clinical relevance the authors analyze and extend published dynamical models of insulin regulated blood glucose homeostasis, develop a small dynamical model of hormone regulated plasma Ca^{2+} homeostasis and discuss other hormone physiological circuits whose structure suggests dynamical compensation as well. Finally, the authors document that the concept of dynamical compensation also applies under pathophysiological conditions during muscle insulin and hepatic insulin resistance.

The study introduces a very interesting concept and succeeds in demonstrating its potential biomedical relevance and is therefore is likely to be of interest to the readers of *Molecular Systems Biology*. We certainly found the paper very interesting to read. There are no issues with the general model. However there is one potentially major uncertainty and potential pitfall in the glucose homeostasis model. The study seems to include the long term effects on functional beta cell mass into a simulation of the short term response towards oral glucose intake. An important question that arises if this is true is: does the predicted functional beta cell mass change during the oral glucose tolerance test? This is not clearly addressed

Major issues:

1) The regulatory component in the glucose homeostasis model that enables dynamic compensation towards insulin sensitivity is the functional mass of beta cells in the pancreas. The amount of secreted insulin is proportional to the functional mass. The beta cell functional mass changes with

respect to the glucose concentration, the higher the glucose concentration the more functional beta cell mass. The authors document such a relationship by referring to a study published in the literature that shows an ultrasensitive dependence of beta cell death on glucose levels, especially around physiological fasting blood glucose levels. This ultrasensitive degradation rate is combined with a production rate that only modestly changes with different glucose levels. Any changes in fasting blood glucose levels that are caused by changes in insulin sensitivity should eventually lead to changes in beta cell mass and therefore insulin secretion. The authors correctly state that such an adaptation is a long term effect of glucose on beta cell mass, although in their model it is simulated as an immediate adaptation on changing glucose levels. The authors need to clarify this point and justify if such a simplification is reasonable, since not only beta cell mass adaptation, but also changes in insulin sensitivity are both long term changes that develop over months.

Nevertheless, beta cell mass is simulated to change during the oral glucose tolerance test. In the model it should increase with increasing glucose levels, while this should not happen in vivo. The mechanism that is proposed to be responsible for the changes of beta cell mass is the combined activity of glucokinase and AMPK, what is reasonable to this reviewer, as long as it describes the long term changes in beta cell mass. The authors show the predicted glucose concentration and the predicted insulin concentration in figure 2. The authors should also show the predicted functional beta cell mass.

A more realistic simulation could first determine a steady state beta cell mass and then simulate the response to oral glucose using this steady state beta cell mass that does not change with changing glucose concentration. How would the simulation look under these conditions?

If not, the authors should discuss their compensatory mechanisms. Insulin secretion is regulated via changing ATP levels in response to glycolysis and thereby by glucokinase, as discussed by the authors. Involvement of AMPK in modulating the acute response to rising glucose levels is not known to this reviewer and reference would be helpful.

2) Supplementary information :

There is some lack of clarity in the relationship between dynamic compensation and pathophysiology.

The glucose tolerance test model should not allow the consideration of insulin insensitivity as the cause for a pathological glucose tolerance test, since it will adapt towards insulin sensitivity with an increase in beta cell mass and higher insulin production to generate the same response. Intuitively, one might assume that a pathological glucose tolerance test is based on a failure of dynamical compensation.

The extended glucose model in suppl. information demonstrates that the concept of dynamical compensation still applies under different insulin sensitivity ratios between muscle and liver. The authors modify their model to distinguish between hepatic and muscle insulin sensitivity and demonstrate that the model still satisfies the requirements for dynamic compensation. This means that the final response, i.e. the increase and decrease of blood glucose levels, depends on the ratio of hepatic and muscle insulin sensitivity, but for any given ratio, the response should always be the same, since it is compensated by changes in beta cell mass. The model's predictions match the clinically measured profiles under primarily hepatic or muscle insulin resistance. The interpretation of these simulated results is that even under pathological conditions dynamical compensation ensures the same response to the same input, though the response itself is not physiological. Why this occurs is not clear. The model should not be able to describe the changes that occur from the physiological state to the pathological state, if the change to pathophysiological state is based on a general loss of insulin sensitivity that cannot be compensated by an increase in beta cell mass and would therefore violate the requirements for dynamical compensation. Are they assuming other mechanisms for pathophysiology? If so the authors should discuss this issue.

The authors do not show the glucose level of a healthy individual in suppl. figure 4a and the measured glucose levels in normal and obese subjects shown in figure 2b are on a different time scale, so it is difficult to compare them with the predictions in suppl. figure 4b. Based on this reviewer's understanding, the simulations that include hepatic and muscle insulin resistance should not be able to predict the physiological oral glucose tolerance test, and the two pathological glucose tolerance tests. Are the parameters for the simulation presented in suppl. figure 2b the same as the parameters used for the simulations discussed in the main text? All missing/different parameters of

the simulation presented in suppl. figure 2b should be presented as well.

Minor issues:

Suppl. figure 2: The ranges for the production rate and glucose-set point are missing.

Supplementary information 3: This section demonstrates that the concept of dynamic compensation also works, if the hormone increases the output parameter. This major difference with the glucose simulation should be mentioned in the main text. The authors show that their model simulates dynamical compensation to secure the plasma calcium level of 1.2mM. In contrast to the beta IG model they do not show the parameters and the source for their parameters. Was the model fitted to reproduce this calcium level?

1st Revision - authors' response

22 September 2016

Thank you very much for the positive consideration of our manuscript and for the reviewer comments. We have now addressed all of the comments in the revised manuscript. We have added a section on ways that DC can break down leading to disease, clarified which aspects of models for other systems are speculative, spelled out the expected timescales of adaptation, improved the definitions and placed the work more precisely in the context of previous work on invariance.

We believe that the revised manuscript is more clear and rigorous.

We detail below the point-by-point changes.

Reviewer 1:

Summary

This paper presents an interesting new viewpoint on how biological systems maintain homeostasis. The novel aspect is the proposal of a class of mechanisms that preserve the temporal characteristics of, say, the response to a pulsed challenge, not just the steady state. They show that standard integral feedback models preserve the latter but not the former, which they term dynamic compensation (DC).

The main application is to glucose homeostasis, and the authors show that an existing class of models developed with that specific application in mind satisfy DC. They also describe more briefly other endocrine systems that have this property, supporting the idea that this is a general principle. Each of these systems responds to short-term fluctuations in the environment by negative feedback and additionally can adapt to chronic failure to meet the target by expanding the mass of endocrine tissue involved.

General Remarks

The work represents a conceptual advance in our understanding of homeostasis, which is not completely characterized by the existence of negative feedback. Although the other models of glucose-insulin homeostasis cited exhibit DC, this feature has not previously been identified properly or discussed, nor has the general principle shared with other endocrine systems been appreciated. This work should be of interest to theorists but also anyone who recognizes the remarkable robustness of homeostatic systems.

We thank the reviewer for this endorsement.

Major comment

l 250: "a circuit with DC is not robust to all parameters": This is a hint that homeostatic failure, leading to diseases such as diabetes, is possible even in systems with DC and would be an opportune place to comment briefly on a key difference between the proposed glucose-insulin model and previous models based on that of Topp et al. The authors' model exhibits DC with respect to insulin sensitivity (S_i), whereas the Topp model can fail to maintain glucose homeostasis if S_i decreases too greatly or too rapidly. This difference results from the choice of functions defining the growth and decay of beta-cell mass, which allows a unique steady state (Supplement, Materials and Methods, l 184 - l 188; Supplementary Figure 2). Whereas the Topp model allows two steady states, one stable,

corresponding to successful homeostasis, and one unstable, which acts as a threshold beyond which mass collapses to a degenerate steady state of 0 mass, common to both models. Because of this, the Topp model exhibits DC provided the stress on the system is not too great but fails if a tipping point is reached.

We thank the reviewer for this comment. In the revised results, we have now added a new section on how DC may fail and on pathways to diabetes.

We now write on page on 9:

”Pathways to failure of DC in glucose homeostasis. Despite its robustness, dynamical compensation fails in some individuals, leading to diseases such as diabetes. Diabetes is characterized by high fasting glucose and impaired glucose dynamics in response to a meal (American Diabetes Association, 2014). Diabetes can occur because of an autoimmune destruction of beta cells (Type 1 Diabetes, T1D) or in a subset of individuals with insulin resistance (Type 2 Diabetes, T2D) or from other reasons. Generally, impaired glucose levels result from insulin secretion that is insufficient given the persons sensitivity to insulin (Bergman et al., 2002).

Topp et al. (Topp et al., 2000) describes three distinct pathways in which such insufficient secretion may develop: regulated hyperglycemia, bifurcation and dynamical hyperglycemia. In regulated hyperglycemia, a change in beta cell removal or production rates causes a change in the glucose set point, such that a hyperglycemic set point is maintained. In bifurcation, a more radical change may cause beta cell removal rate to exceed beta cell production rate at all glucose concentrations, resulting in the elimination of the beta cell population, which may occur in T1D. The third pathway, dynamical hyperglycemia, relies on the existence of an unstable fixed point at a high glucose concentration due to the toxic effect of glucose on beta cells at these concentrations (Efanova et al., 1998). Topp et al. show that in this case, if insulin sensitivity drops faster than beta cell functional mass can adapt, then glucose levels may exceed this unstable fixed point. In this case, the beta cell population is eliminated. This pathway may underlie the etiology of T2D (Ha et al., 2015).

The above three pathways result in a perturbed glucose steady state level. Hence, one of the conditions for DC is not met, condition (i) (stability at the desired set point). We would like to add two other mechanisms for pathology that can occur even when the normal glucose set point is maintained. First, note that a circuit with DC is not robust to all of its parameters, only to certain ones. The glucose homeostasis model has DC to the insulin sensitivity and insulin secretion parameters, which vary over a wide range. The model does not have, by itself, DC to variation in endogenous glucose production or insulin removal rate, which may vary less. Changes in these parameters alter glucose dynamics in response to a given input (Extended Data Fig. 3 and Supplementary Information 4). DC in the model is also affected by a mismatch between muscle and liver insulin resistance, which can alter glucose dynamics in a way that agrees with clinical observations (Extended Data Fig. 4 and Supplementary Information 5). “

Minor Points

l 101: the result $\hat{x}_{st} = u_0$ assumes that $u(t)$ is 0

corrected

l 112: g has an extra argument, 0

corrected

l 142: The beta-I-G model: this name has not been defined

corrected

Cross-commenting remarks:

I don't agree with Reviewer #2's insistence that the concept of dynamical compensation needs to be demonstrated with a more realistic or complex model. That would be tantamount to saying that Newton's law of gravity was not valid because it was based on observations of the oversimplified case of a falling apple instead of, say, the flight of a bird. This review is a classic case of "you didn't write the paper I would have written." It is reasonable to ask the authors to clarify how their new concept compares to other related ideas, such as fold detection, but it is not reasonable to ask them to set aside their findings until they complete a research program that could take years.

Reviewer #3 is confused about the simulations in the paper. The authors have not allowed beta cell mass to change during the course of an OGTT; they have done what he suggests they should do. This is a minor point that the authors can address easily in their rebuttal.

I do agree with the requests of both reviewers for more complete specification of the models used, parameters, etc. I stand by my recommendation that the paper can be made acceptable by minor revisions.

Reviewer 2:

Summary

The paper introduces the concept of Dynamical Compensation (DC), a seemingly novel property of some negative feedback loops involving an integral action (or what the authors call a slow feedback). In addition to the perfect adaptation property with respect to a given input stimulus, the dynamical compensation property ensures that, for some given input stimulus, the regulated variable will have the same response independently of the value of certain network parameters. It is a very strong property that can be understood as an invariance property with respect to some network parameters which is reminiscent of the fold-change detection property [1], an invariance property with respect to constant scaling of constant inputs.

Numerical comparisons (on not necessarily physiologically realistic models) between different integral feedback strategies illustrate that this property is not ensured by every integral feedback motifs. A result stating simple sufficient conditions for a 3-node network (involving an integral feedback) to exhibit this property is also proved. Based on work reported in the literature, the authors show that a simple model of glucose homeostasis possesses such a property, and they exploit it to make predictions that seem to be validated by some existing experimental data. Finally, the authors extrapolate this result to a more complex glucose homeostasis system and to other physiological models (such as calcium regulation), this time with little supporting evidence.

General remarks

The paper is essentially of theoretical nature, as all the experimental data are taken from the existing literature in the field. Also, the paper makes strong claims that are not supported by any experimental data or experimental evidence. Nevertheless, the theoretical concept of dynamical compensation is very interesting and should have been investigated in more detail. Indeed, the results are very simple and lack generality, as they only apply to toy models. Also the connection with invariance should have been made clearer and studied in more detail using, for instance, similar ideas as in [1]. Indeed, even if fold-change detection is a different phenomenon, they are both invariance properties. In my opinion, the name "dynamical compensation" is not very suitable, as it does not properly describe the reported property. "Dynamical invariance" or "parametric invariance" seem more appropriate. In spite of this, this work nicely complements existing work pertaining to the integral control of reaction networks (such as [1]) which have clearly missed this property. Overall, I believe the paper makes a nice conceptual advance and may be publishable, but only after major revisions have been made.

We thank the reviewer for this endorsement.

Major Comments

1 One of the main concerns with this work is its lack of generality. First of all, the authors only consider 3-node topologies (as e.g. in [2]), which is quite restrictive. The first question that comes to mind is whether this property may hold for larger networks and, if so, how can we check it?

We have now added evidence that a certain relevant class of larger networks also has the DC property. This class arises in the case of insulin, where insulin passes through several compartments in order to affect target tissues. These compartments are usually modeled as a series of linear equations, $dI_k/dt = \eta_k I_{k-1} - \mu_k I_k$. We show that this series of compartments still preserves the DC property in the revised SI section 1.

We also added to the discussion a call for future work to analyze larger networks:

“Our analysis of circuits with the DC property focused on circuits with three nodes. A three-node circuit architecture allows for a fast feedback component together with a slower nonlinear integral feedback component, which may correspond in physiological systems to a hormone and a hormone-secreting tissue, respectively. Future work may explore dynamical compensation in more complex regulatory networks.”

Another limitation (which is also present in [2]) is that the input stimulus is only applied to one specific node, but what would happen if it were applied to another one? Technically, input stimuli may perturb the network at every possible node and reaction.

While theoretically any node may be externally perturbed, this seems unrealistic for the systems that we discuss in the paper.

2 The different illustrative toy models have been often badly chosen. Indeed, some of their states can go negative, which is not very realistic from a physiological viewpoint. For instance, on page 3, a much better model could have been used than the one considered. Indeed, the model

$$\begin{aligned}y'(t) &= u_0 + u(t) - sx(t)y(t) \\x'(t) &= pZ(t) - x(t) \quad (1) \\Z(t) &= Z(t)(y(t) - y_0)\end{aligned}$$

would also have the DC property, would be simpler and would have its state confined to the nonnegative orthant. Additionally, the fact that the term pyZ has been changed to pZ makes the model more interesting since x now represents the variable that will, in turn, activate the degradation of the controlled variable y . Note also that neither this model nor the original one does include proportional-integral feedback. These models only involve an integral one since the dynamics of x cannot be considered as a proportional action. It is simply low-pass filtering of the integral action. The models given in Figure 1 should also be modified so that they become physiologically more relevant. Note that this modification is possible while still keeping the main point of the authors: that not all networks with integral feedback exhibit the DC property.

We thank the reviewer for this comment. We have now incorporated in Figure 1 and in page 4 a DC model that is physiologically feasible and is confined to the nonnegative orthant as suggested. The model reads:

“We propose a mechanism for DC based on known hormonal circuit reactions (Fig 1d). The basic idea is that the regulated variable y controls the functional mass Z of the tissue that secretes the hormone x that regulates y . The feedback gain of x is s and the feedback gain of Z is p and the circuit input is $u(t)$. The circuit dynamic equations are as follows:

$$y' = u_0 + u(t) - sxy \quad [1]$$

$$x' = pZy - x \quad [2]$$

$$Z' = Z \cdot (y - y_0) \quad [3]$$

This circuit describes nonlinear integral feedback on the level of y . The nonlinearity of Eq. [3] stems from the fact that Z are cells, and hence their growth equation is autocatalytic $Z' = Z\alpha$ where α is the growth rate. In this case α depends on the regulated variable y such that growth is zero when $y = y_0$. For example, y can increase the proliferation rate λ_+ and/or decrease the removal rate λ_- of cells, such that the two rates cross at $y = y_0$ and the growth rate is $\alpha = \lambda_+ - \lambda_-$.“

3. Another point that needs to be addressed is the extent to which this property is robust. From a biological viewpoint, fragile functions or properties are not achievable. A discussion is necessary to explain how robust is the DC property. In other words, when a model is assumed to satisfy the DC property, to what extent a perturbed version of that model also satisfies that property? For instance, if a degradation term is added to the integrator (for instance, the term $-\delta Z$ where $\delta > 0$), then the adaptation property as well as the DC property are lost. These properties are non-robust with respect to perturbations at the integrator level, which is not surprising. Robustness is also essential to consider in order to capture the fact that biological systems are not isolated from each other but interfere in sometimes very strong ways. In the present case, the α -cells also act on the glucose concentration. In summary, the robustness of this property and that of the glucose regulation system should be addressed and properly discussed.

We thank the reviewer for this comment. We now address fragility in the context of potential failure of DC in diabetes in a new section on page 9. The case mentioned by the referee, a degradation term added to the integrator (for instance, the term $-\delta Z$ where $\delta > 0$), leads to an effective shift of δ in the glucose fixed point y_0 . This corresponds to regulated hyperglycemia or bifurcation mechanisms in the terminology of Topp et al. We address several other mechanisms for failure in this section:

“Pathways to failure of DC in glucose homeostasis. Despite its robustness, dynamical compensation fails in some individuals, leading to diseases such as diabetes. Diabetes is characterized by high fasting glucose and impaired glucose dynamics in response to a meal (American Diabetes Association, 2014). Diabetes can occur because of an autoimmune destruction of beta cells (Type 1 Diabetes, T1D) or in a subset of individuals with insulin resistance (Type 2 Diabetes, T2D) or from other reasons. Generally, impaired glucose levels result from insulin secretion that is insufficient given the persons sensitivity to insulin (Bergman et al., 2002). Topp et al. (Topp et al., 2000) describes three distinct pathways in which such insufficient secretion may develop: regulated hyperglycemia, bifurcation and dynamical hyperglycemia. In regulated hyperglycemia, a change in beta cell removal or production rates causes a change in the glucose set point, such that a hyperglycemic set point is maintained. In bifurcation, a more radical change may cause beta cell removal rate to exceed beta cell production rate at all glucose concentrations, resulting in the elimination of the beta cell population, which may occur in T1D. The third pathway, dynamical hyperglycemia, relies on the existence of an unstable fixed point at a high glucose concentration due to the toxic effect of glucose on beta cells at these concentrations (Efanova et al., 1998). Topp et al. show that in this case, if insulin sensitivity drops faster than beta cell functional mass can adapt, then glucose levels may exceed this unstable fixed point. In this case, the beta cell population is eliminated. This pathway may underlie the etiology of T2D (Ha et al., 2015). The above three pathways result in a perturbed glucose steady state level. Hence, one of the conditions for DC is not met, condition (i) (stability at the desired set point). We would like to add two other mechanisms for pathology that can occur even when the normal glucose set point is maintained. First, note that a circuit with DC is not robust to all of its parameters, only to certain ones. The glucose homeostasis model has DC to the insulin sensitivity and insulin secretion parameters, which vary over a wide range. The model does not have, by itself, DC to variation in endogenous glucose production or insulin removal rate, which may vary less. Changes in these parameters alter glucose dynamics in response to a given input (Extended Data Fig. 3 and Supplementary Information 4). DC in the model is also affected by a mismatch between muscle and liver insulin resistance, which can alter glucose dynamics in a way that agrees with clinical observations (Extended Data Fig. 4 and Supplementary Information 5). “

4. Page 8. It is claimed in the main text that by adding a slow feedback in the model taken from Dalla Man et al., 2007, the model then exhibits DC. This should be clearly developed in the supplementary material. In the current version of the paper, nothing is explained or even proved. The model needs to be included for completeness, and the conditions of the main result should be applied to it in order to prove the DC property.

We added an SI item (Supplementary information 2) that provides an overview of the Dalla Man model. Due to the very large scope of the model, we refer the reader to the original paper for complete details. We also note that our study of this model with and without the new slow feedback loop was done numerically; because of its complexity we did not attempt an analytical proof.

5. The authors assume in the calcium regulation the presence of a slow feedback implementing an integral action where the functional mass of the parathyroid gland is modeled as $M' = M \cdot h([Ca^{2+}])$. However, no evidence is given that this is indeed the case. The authors do not cite any experimental paper that supports such a model.

We thank the reviewer for this comment. We now added experimental evidence to the relevant SI section on page 8. The evidence is as follows:

“As was the case with beta cell mass, we assume here that the functional mass of the parathyroid gland is chiefly controlled plasma calcium. This assumption is supported by several experimental studies. Low calcium diet causes a 10-fold increase in PT-cell proliferation in rodents (Naveh-Many et al., 1995), while direct activation of the calcium receptor inhibits PT-cell proliferation (Chen, 2004; Chin et al., 2000; Wada, 2003; Wada et al., 1997, 2000) and increases PT-cell apoptosis (Mizobuchi et al., 2007). “

Moreover, dynamical compensation does not seem to have been reported in the case of calcium homeostasis.

We have added a statement on page 10 that experimental data is lacking on dynamical compensation in the circuits described. We clarify that the additional systems we mention (other than glucose homeostasis) have experimental evidence for the slow feedback loop on tissue mass by the regulated variable. We clarify that they are only candidates for DC because they lack direct experimental evidence for the DC property.

Thus, this example seems quite contrived. One can always "fix" a model in order to have some desired properties, but this can only be considered as speculative.

We believe that the experimental evidence mentioned above makes this case more plausible.

6. Regarding Section 5 of the SI, the glucose dynamics can go negative and this should be corrected. The model also needs better justification.

The reviewer is correct to point out that glucose can get negative values in the model presented. However, we decided to keep this model because of its importance - it is based on detailed models that are used for analyzing clinical data (e.g <https://www.ncbi.nlm.nih.gov/pmc/articles/PMC2584819/>). We have added justification for the model in the first paragraph of this SI section (SI page 9).

7. All the figures in the supplementary material need additional explanations/interpretations in the text or in their legend. In particular, Figure 2a and Figure 4 need further explanations.

We added explanations for all of the SI figures.

8. In Figure 2, the experimental curves are averaged over several individuals, a procedure that both eliminates the randomness in the trajectories and the fact that these trajectories can be very different from each other. Nothing is said about this. A consequence is that fitting a model on these average trajectories may mean nothing since the deterministic model would represent an individual while the used data is a population average. Hence, the considered model seems to be more a phenomenological model than a mechanistic one, which goes against all the arguments used in the paper to justify the structure of the models. This should be clarified.

We now mention this caveat in the discussion on page 13:

"DC seems to occur in experimental measurements on the glucose and insulin responses of people with and without insulin resistance (Fig 2, insets). These studies reported population averages of the dynamics, which can potentially mask variations between people; data on individual dynamics would provide a more stringent test of DC."

9. It is said in the discussion section that DC is different from FCD. While this is true, the result presented in the paper is also an invariance property and should have been addressed as such using the same ideas as in [1]. It is important that the paper is rewritten to account for this similarity.

We now guide the reader to the invariance property in the discussion, and refer to Ref [1].

We write:

"The concept of dynamical compensation relates to the concept of fold change detection (FCD), in which the output dynamics of a system is independent of multiplying its input by a scalar (Shoval et al., 2010). Like FCD, which is an invariance property (Shoval et al., 2011) with respect to scaling of the input, dynamical compensation is also an invariance property, but with respect to changes in certain parameters. DC, however, is different from FCD because most known FCD mechanisms do not have dynamical compensation when their parameters are changed. Likewise, DC systems need not have FCD."

Minor points

1. In the notations, it is important to make the distinction between initial conditions (with subscript 0, like x_0) and steady-state values with superscript * for instance like x^* . For instance, in the integrator expression, y should be used.

We now use a superscript * in the statement and proof of the general theorem.

1 Page 3. The definition of DC is very sloppy. It should be formulated in more rigorous way (i.e. in a mathematically rigorous way) with a possible explanation that could be similar to the one given in the present version.

The definition of DC was formulated in page 3:

”Definition of dynamical compensation. Consider a system with an input $u(t)$ and an output $y(t,s)$ such that $s > 0$ is a parameter of the system. The system is initially at steady state with $u(0) = 0$. Dynamical compensation (DC) with respect to s is that for any input $u(t)$ and any (constant) s the output of the system $y(t,s)$ does not depend on s . That is, for any s_1, s_2 and for any time dependent input $u(t)$ then $y(t, s_1) = y(t, s_2)$.”

2 Page 5. The statements of the conditions should also be reformulated in a less sloppy way. For instance, the uniqueness of the equilibrium point and the local stability could be stated together. The homogeneity property can be stated on a separate item.

Conditions of the general theorem were reformulated in page 5 of the main text and in page 2 of the SI.

“Sufficient conditions for dynamical compensation. A more general class of models that show DC with respect to variations in their parameters p, s is: $y = f(u, y, sx)$, $x = g(y, pZ, x)$, $Z = Z \cdot h(y)$, provided the following sufficiency conditions: (i) For all p, s , the system is stable at $y = y^*$, there exists a unique solution $sx = x^*$ for $f(0, y^*, sx) = 0$ and there exists a unique solution $psZ = Z^*$ for $g(y^*, psZ, x^*) = 0$ (ii) A factorization condition on the function g : $g(y, psZ, sx) = sg(y, pZ, x)$. Proof in (Supplementary Information 1). This model also extends naturally and shows DC when x passes through multiple compartments (Supplementary Information 1).”

3 The statement and the proof of the main theoretical result need improvement. The first statement should be about the existence, uniqueness, and stability of the equilibrium point for all $s, p > 0$. The second statement is the factorization condition. Note that homogeneity is usually defined in the case of a scaling factor that applies to all the arguments of a function. This is not the case here. Hence, I would recommend to change the term to avoid confusion. The term factorization seems more appropriate. In the proof, notation using equal signs in the arguments of a function should be avoided.

We now improved this as suggested. Conditions of the general theorem were changed in page 5 of the main text and page 2 of the SI.

4 It should be discussed how this property generalizes to larger models (more than 3 nodes) **A discussion of larger networks was added as discussed above.** and to other stimuli, like sudden change in the parameters of the system. For instance, what happens if s or p suddenly change while u is kept constant? **This seems to be outside the scope of this paper.**

5 It seems that this property crucially depends on structure of the integral action. What would happen in the presence of a small leakage in the dynamics of Z or the consideration of another integral control structure?

We discussed the dependency of the property on the dynamics of Z in the context of the pathways to diabetes in page 9. Considering another integral control structure seems to be outside the scope of this work.

Reviewer three

In this study the authors introduce the concept of dynamical compensation and explain its biomedical relevance using hormone regulatory circuits as examples. Dynamical compensation describes the ability of a biological system to respond to the same input always with the same dynamical output regardless of the actual states of some of its parameters. The authors explain and provide proof the concept first by using a simple example of a dynamical model that contains the necessary network motif, followed by a generalized model description and proof. To demonstrate the biological/clinical relevance the authors analyze and extend published dynamical models of insulin regulated blood glucose homeostasis, develop a small dynamical model of hormone regulated plasma Ca^{2+} homeostasis and discuss other hormone physiological circuits whose

structure suggests dynamical compensation as well. Finally, the authors document that the concept of dynamical compensation also applies under pathophysiological conditions during muscle insulin and hepatic insulin resistance.

The study introduces a very interesting concept and succeeds in demonstrating its potential biomedical relevance and is therefore likely to be of interest to the readers of Molecular Systems Biology. We certainly found the paper very interesting to read.

We thank the reviewer for this endorsement.

There are no issues with the general model. However there is one potentially major uncertainty and potential pitfall in the glucose homeostasis model. The study seems to include the long term effects on functional beta cell mass into a simulation of the short term response towards oral glucose intake. An important question that arises if this is true is: does the predicted functional beta cell mass change during the oral glucose tolerance test? This is not clearly addressed

Major issues:

1) The regulatory component in the glucose homeostasis model that enables dynamic compensation towards insulin sensitivity is the functional mass of beta cells in the pancreas. The amount of secreted insulin is proportional to the functional mass. The beta cell functional mass changes with respect to the glucose concentration, the higher the glucose concentration the more functional beta cell mass. The authors document such a relationship by referring to a study published in the literature that shows an ultrasensitive dependence of beta cell death on glucose levels, especially around physiological fasting blood glucose levels. This ultrasensitive degradation rate is combined with a production rate that only modestly changes with different glucose levels. Any changes in fasting blood glucose levels that are caused by changes in insulin sensitivity should eventually lead to changes in beta cell mass and therefore insulin secretion. The authors correctly state that such an adaptation is a long term effect of glucose on beta cell mass, although in their model it is simulated as an immediate adaptation on changing glucose levels. The authors need to clarify this point and justify if such a simplification is reasonable, since not only beta cell mass adaptation, but also changes in insulin sensitivity are both long term changes that develop over months.

Nevertheless, beta cell mass is simulated to change during the oral glucose tolerance test. In the model it should increase with increasing glucose levels, while this should not happen in vivo. The mechanism that is proposed to be responsible for the changes of beta cell mass is the combined activity of glucokinase and AMPK, what is reasonable to this reviewer, as long as it describes the long term changes in beta cell mass. The authors show the predicted glucose concentration and the predicted insulin concentration in figure 2. The authors should also show the predicted functional beta cell mass. A more realistic simulation could first determine a steady state beta cell mass and then simulate the response to oral glucose using this steady state beta cell mass that does not change with changing glucose concentration. How would the simulation look under these conditions?

If not, the authors should discuss their compensatory mechanisms. Insulin secretion is regulated via changing ATP levels in response to glycolysis and thereby by glucokinase, as discussed by the authors. Involvement of AMPK in modulating the acute response to rising glucose levels is not known to this reviewer and reference would be helpful.

We thank the reviewer for this comment, which helped us to clarify an important point. We now discuss the timescales in the insulin/glucose system. The rate of change of beta cell mass is much slower than the acute response timescale. It is expected to be on the scale of days to months. Therefore, on the timescale of a meal or an oral glucose tolerance test, beta cell mass does not appreciably change in the model. Upon a step-like change in insulin sensitivity, therefore, the model shows a transient period of days to months in which dynamics will be abnormal. Only after beta cells reach their new steady state that precisely compensates for the change in insulin sensitivity, will the postprandial glucose dynamics return to their normal, pre-change level. As the referee notes, more realistic changes in insulin sensitivity are spread over months. In this case, beta cell dynamics will be able to track the change in S_i and effectively compensate the dynamics throughout.

We now discuss this in the results section on page 7:

“Note that the adaptation of beta cell functional mass to the change in insulin sensitivity may take several days to months, and only after adaptation are the glucose dynamics precise. Therefore, after a step-like change in insulin sensitivity the model shows a period of time in which glucose dynamics are not fully compensated. Upon changes in insulin sensitivity that occur gradually over months, the beta cells in the model will be able to track the changes in S_i and effectively compensate glucose dynamics throughout.”

2) Supplementary information :

2) There is some lack of clarity in the relationship between dynamic compensation and pathophysiology.

The glucose tolerance test model should not allow the consideration of insulin insensitivity as the cause for a pathological glucose tolerance test, since it will adapt towards insulin sensitivity with an increase in beta cell mass and higher insulin production to generate the same response. Intuitively, one might assume that a pathological glucose tolerance test is based on a failure of dynamical compensation.

The extended glucose model in suppl. information demonstrates that the concept of dynamical compensation still applies under different insulin sensitivity ratios between muscle and liver. The authors modify their model to distinguish between hepatic and muscle insulin sensitivity and demonstrate that the model still satisfies the requirements for dynamic compensation. This means that the final response, i.e. the increase and decrease of blood glucose levels, depends on the ratio of hepatic and muscle insulin sensitivity, but for any given ratio, the response should always be the same, since it is compensated by changes in beta cell mass. The model's predictions match the clinically measured profiles under primarily hepatic or muscle insulin resistance. The interpretation of these simulated results is that even under pathological conditions dynamical compensation ensures the same response to the same input, though the response itself is not physiological. Why this occurs is not clear. The model should not be able to describe the changes that occur from the physiological state to the pathological state, if the change to pathophysiological state is based on a general loss of insulin sensitivity that cannot be compensated by an increase in beta cell mass and would therefore violate the requirements for dynamical compensation. Are they assuming other mechanisms for pathophysiology? If so the authors should discuss this issue.

We thank the reviewer for this comment, which helped us to add a new section of failure of DC in the context of glucose homeostasis. We believe that this increases the relevance of this study in suggesting putative mechanisms for pathophysiology.

The new section in the results on page 9 reads:

“Pathways to failure of DC in glucose homeostasis. Despite its robustness, dynamical compensation fails in some individuals, leading to diseases such as diabetes. Diabetes is characterized by high fasting glucose and impaired glucose dynamics in response to a meal (American Diabetes Association, 2014). Diabetes can occur because of an autoimmune destruction of beta cells (Type 1 Diabetes, T1D) or in a subset of individuals with insulin resistance (Type 2 Diabetes, T2D) or from other reasons. Generally, impaired glucose levels result from insulin secretion that is insufficient given the person's sensitivity to insulin (Bergman et al., 2002). Topp et al. (Topp et al., 2000) describes three distinct pathways in which such insufficient secretion may develop: regulated hyperglycemia, bifurcation and dynamical hyperglycemia. In regulated hyperglycemia, a change in beta cell removal or production rates causes a change in the glucose set point, such that a hyperglycemic set point is maintained. In bifurcation, a more radical change may cause beta cell removal rate to exceed beta cell production rate at all glucose concentrations, resulting in the elimination of the beta cell population, which may occur in T1D. The third pathway, dynamical hyperglycemia, relies on the existence of an unstable fixed point at a high glucose concentration due to the toxic effect of glucose on beta cells at these concentrations (Efanova et al., 1998). Topp et al. show that in this case, if insulin sensitivity drops faster than beta cell functional mass can adapt, then glucose levels may exceed this unstable fixed point. In this case, the beta cell population is eliminated. This pathway may underlie the etiology of T2D (Ha et al., 2015). The above three pathways result in a perturbed glucose steady state level. Hence, one of the conditions for DC is not met, condition (i) (stability at the desired set point). We would like to add two other mechanisms for pathology that can occur even when the normal glucose set point is

maintained. First, note that a circuit with DC is not robust to all of its parameters, only to certain ones. The glucose homeostasis model has DC to the insulin sensitivity and insulin secretion parameters, which vary over a wide range. The model does not have, by itself, DC to variation in endogenous glucose production or insulin removal rate, which may vary less. Changes in these parameters alter glucose dynamics in response to a given input (Extended Data Fig. 3 and Supplementary Information 4). DC in the model is also affected by a mismatch between muscle and liver insulin resistance, which can alter glucose dynamics in a way that agrees with clinical observations (Extended Data Fig. 4 and Supplementary Information 5). ”

3) The authors do not show the glucose level of a healthy individual in suppl. figure 4a and the measured glucose levels in normal and obese subjects shown in figure 2b are on a different time scale, so it is difficult to compare them with the predictions in suppl. figure 4b.

We now added a simulation of a healthy/normal glucose response to supplementary figure 4b.

Based on this reviewers understanding, the simulations that include hepatic and muscle insulin resistance should not be able to predict the physiological oral glucose tolerance test, and the two pathological glucose tolerance tests. Are the parameters for the simulation presented in suppl. figure 2b the same as the parameters used for the simulations discussed in the main text? All missing/different parameters of the simulation presented in suppl. figure 2b should be presented as well.

The parameters for this figure were added to the methods section of the SI. We also made clear how the simulations of figure 4b were generated in page 7 of the SI. Note that they were not explicitly created to fit the data of figure 4a, but instead represent the most extreme case of complete resistance in either muscle or liver insulin sensitivity in our minimal model.

Minor issues:

1) Suppl. figure 2: The ranges for the production rate and glucose-set point are missing. We corrected the figure to include the ranges for production rate and glucose set point.

2) Supplementary information 3: This section demonstrates that the concept of dynamic compensation also works, if the hormone increases the output parameter. This major difference with the glucose simulation should be mentioned in the main text.

We now added this to the revised results:

“This model demonstrates that DC can occur also when the hormone acts to increase the regulated variable (calcium), and not only when it acts to decrease it as in the case of glucose/insulin.”

The authors show that their model simulates dynamical compensation to secure the plasma calcium level of 1.2mM. In contrast to the beta IG model they do not show the parameters and the source for their parameters. Was the model fitted to reproduce this calcium level?

We added to page 10 a clarification that there is insufficient experimental evidence on whether or not indeed there is DC in the calcium homeostasis system. Nevertheless there is substantial experimental evidence that calcium controls parathyroid mass dynamics, and we added several references on this to page 9 of the SI. There are currently no accurate measurements on the exact dependence of parathyroid mass dynamics on calcium, so we cannot determine that indeed the parathyroid is stable at this concentration and hence our model is based on documented interactions but is hypothetical on the precise parameters. We rephrased several sentences in page 9 of the SI and page 10 of the main text to make this more clear.

In summary, the reviewer comments helped us improve the relevance of the paper to pathophysiology, increase the rigor and clarity with respect to timescales, better situate the manuscript with respect to invariance properties and to clarify which aspects of the models are documented versus speculative. We believe that the revised paper is much improved and we thank the reviewers for their excellent comments.

2nd Editorial Decision

05 October 2016

Thank you again for sending us your revised manuscript. We are now satisfied with the modifications made and I am pleased to inform you that your paper has been accepted for publication.

Corresponding Author Name: Prof. Uri Alon

Manuscript Number: MSB-16-7216